# Artificial Neural Network Prediction of Antiadhesion and Antibiofilm-Forming Effects of Antimicrobial Active Mushroom Extracts on Food-Borne Pathogens

**DOI:** 10.3390/antibiotics12030627

**Published:** 2023-03-22

**Authors:** Jovana Vunduk, Anita Klaus, Vesna Lazić, Maja Kozarski, Danka Radić, Olja Šovljanski, Lato Pezo

**Affiliations:** 1Institute of General and Physical Chemistry, Studenski trg 10-12, 11 158 Belgrade, Serbia; 2Institute for Food Technology and Biochemistry, Faculty of Agriculture, University of Belgrade, Nemanjina 6, 11 080 Belgrade, Serbia; 3Faculty of Technology Novi Sad, University of Novi Sad, Bulevar Cara Lazara 1, 21 000 Novi Sad, Serbia

**Keywords:** antiadhesion, antibiofilm, food-borne pathogens, mushroom extracts, artificial neural network, model

## Abstract

The problem of microbial biofilms has come to the fore alongside food, pharmaceutical, and healthcare industrialization. The development of new antibiofilm products has become urgent, but it includes bioprospecting and is time and money-consuming. Contemporary efforts are directed at the pursuit of effective compounds of natural origin, also known as “green” agents. Mushrooms appear to be a possible new source of antibiofilm compounds, as has been demonstrated recently. The existing modeling methods are directed toward predicting bacterial biofilm formation, not in the presence of antibiofilm materials. Moreover, the modeling is almost exclusively targeted at biofilms in healthcare, while modeling related to the food industry remains under-researched. The present study applied an Artificial Neural Network (ANN) model to analyze the anti-adhesion and anti-biofilm-forming effects of 40 extracts from 20 mushroom species against two very important food-borne bacterial species for food and food-related industries—*Listeria monocytogenes* and *Salmonella enteritidis*. The models developed in this study exhibited high prediction quality, as indicated by high r^2^ values during the training cycle. The best fit between the modeled and measured values was observed for the inhibition of adhesion. This study provides a valuable contribution to the field, supporting industrial settings during the initial stage of biofilm formation, when these communities are the most vulnerable, and promoting innovative and improved safety management.

## 1. Introduction

In December 2021 the Dole company from the USA recalled its iceberg lettuce and any finished products containing it. A Food and Drug Administration (FDA) and Center for Disease Control (CDC) investigation showed that 18 people from 13 states ended up infected with *L. monocytogenes* found in Dole’s salad [1]. The traces led to the harvesting equipment. The same company has been involved in *L. monocytogenes* outbreaks in the past because the biofilm this food-borne pathogen produces enables it to stick to processing equipment. Hudson Valley Farms recalled their nuts and nut butter products because of the same problem of *Listeria* biofilm [2]. Although both companies practiced HACCP, which recognizes bacteria-connected critical points, this safety system does not necessarily include biofilm prevention measures or eradication. Similarly, most of the research focuses on species that form biofilms within the food industry, leaving the associated biofilm out of the investigations [3]. It has been pointed out that there is an existing gap between industrial practices and academic research on biofilms [4]. The focus has been mainly on fundamental knowledge regarding biofilms, while applicable solutions (beyond physicochemical measures) for the industry are lacking. Biofilms are a common enemy across the food industry; they can be encountered in dairy, meat, ready-to-eat food, baby food, water, juices, or product manufacturing facilities. Nutrients; moisture; hard-to-reach, clean, and sanitize areas; and cross-contamination enable the growth of pathogenic bacteria followed by their adhesion to surfaces, biofilm-formation, maturation, and dispersal [5]. One outcome of this is bacterial infections, intoxications, food safety and quality deterioration, and manufacturing equipment corrosion [6]. The most common sanitation measures include different chemical agents such as chlorine-based sanitizers, hydrogen peroxide, ozone, quaternary ammonium compounds, salicylate-based polyanhydride esters, and synthetic brominated furanone [7]. However, they are not effective enough at all times because bacteria develop resistance. The presence of biofilms within meat processing facilities has been assessed previously, and it has been discovered that among 108 samples collected during processing and after cleaning and disinfection, 10 were biofilm hotspots [3]. Three originated after the cleaning and disinfection. This proves that innovation is urgently needed in this sector. It has also been emphasized that society needs to move toward “green” antibiofilm agents because conventional methods of biofilm control can cause the emergence of resistance and are costly and have low efficiency, while at the same time they can have a negative impact on human health and the environment [8]. Novel approaches consider the use of bacteriophages, bacteriocins, enzymes, essential oils, and substances known as quorum-sensing inhibitors [9]. There are even fewer compounds that can inhibit biofilm formation without exhibiting antimicrobial activity at the same time. Among substances of natural origin, most are phytochemicals such as carvacrol, hordenin, ginkgolic acid, epigallocatechin-3-gallate, isosteviol, phytol, curcumin, tannic acid, and 3,3′-diindolylmethane, to name a few [10,11,12].

Plants are already well-known and traditionally used sources of antimicrobial and antibiofilm compounds. Different parts, such as leaf, root, fruit, or peel, are rich in antibiofilm compounds, as has been demonstrated recently using the simple and cheap crystal violet staining method [13]. Recently, mushrooms became an interesting source of antimicrobial compounds, though only a few papers have dealt with the antibiofilm activity of mushroom-derived agents [14]. A pioneer work dealing with several methanol–water wild mushroom extracts against biofilms of selected clinical isolates of pathogenic bacteria was published in 2014 [15]. All tested samples exhibited some degree of antibiofilm activity, while at the same time they expressed no cytotoxicity on porcine liver cells. Another group of authors examined aqueous extracts of wild mushrooms and reported antiadhesion activity in the range of 26.6 to 73.7% against several pathogenic bacteria and fungi [16]. Two different groups of researchers isolated several low-molecular-mass compounds from shiitake mushroom and demonstrated their antibacterial and antibiofilm activity (mainly in the adhesion phase) against bacteria involved in caries and gingivitis [17,18]. When it comes to more commercially sustainable materials of mushroom origin, such as mycelium, Vunduk et al. [19] applied submerged cultivation technology to obtain exo- and endo-polysaccharides from *Pleurotus flabellatus* strain Mynuk, which expressed antiadhesion and antibiofilm activity against ATCC and the clinical isolates of several pathogens. Chlorinated orcinol derivate, a compound from the mycelial culture of North American *Hericium* sp. has been demonstrated as an effective agent in the prevention of pathogenic fungi biofilm formation [20]. One should remember that to be applied in the food industry, the substance must not be toxic or poorly soluble [21]. Thus, materials such as polysaccharides from mushrooms, which have been proven as non-cytotoxic against normal human cell lines, whose extracts are water-soluble, or solubility can be improved, are promising antibiofilm agents for the food industry. 

As recently cited here, there are several promising compounds of natural origin, and only a scarce number of them have been tested in a clinical setting. According to the database of [22], only a frankincense extract (derived from *Boswellia sacra* Flueck.) got to the clinical trial phase for periodontal diseases [23]. This implies that the path from initial screening to commercial product is long, requires robust analysis, and is costly. On the other hand, the industry needs easy-to-apply, cheap, and innovative solutions. As an outcome of a series of academic–industry knowledge exchange workshops, a number of key challenges have been identified, and improved biofilm analysis methods including artificial intelligence and machine learning are one of them [6]. Shaban and Alkawareek [24] presented a machine learning-based predictive modeling approach for a qualitative prediction of the in vitro antibiofilm activity of antibiotics. However, there are no studies examining the other machine learning options, as well as those for mushroom-derived compounds, which are of interest to the food industry. 

Thus, the goal of this study was to apply ANN as a good mathematical tool when there is a high variability of the variables and the nonlinear behavior of process parameters [25], which are all characteristics of bacterial biofilms. Furthermore, ANN systems are successively used in a novel study of the antimicrobial potential of natural-based agents with a high possibility of predicting all connected characteristics of the antimicrobial pathways [26,27,28].

It should be of significant use in the industrial setting, such as food manufacturing, which cannot deal with time- and resources-consuming assays. The ANN model does not need physical model parameters, but still encompasses a capacity to obtain results from the experimental data and is able to handle the complex system with nonlinearities and to elaborate the interactions between variables [29,30]. 

## 2. Results

Considering the aim of testing, a high number of different mushroom extracts was involved in antimicrobial and antibiofilm activity assays; the list of mushroom species, specific extracts, and their abbreviations are given in Appendix A.

### 2.1. Antimicrobial Activity

The majority of tested samples exhibited antimicrobial activity toward both strains of food-borne bacteria, with minimal inhibitory concentrations (MIC) being mainly in the range of 10–20 mg/mL (Table 1). Specific sensitivity to mushroom extracts was not observed according to Gram-positive or Gram-negative bacteria. However, none of the samples showed a bactericidal effect when incubated with *S. enteritidis*, while the alkali extract of *Meripilus giganteus* and *Ganoderma lucidum* killed Gram-positive *L. monocytogenes* when the maximal tested concentration (20 mg/mL) was applied. In all cases, except one, the MIC of alkali-extracted samples was lower than the MIC of water extracts. In several cases, there was no difference concerning the method of extraction and antibacterial effect. The strongest inhibition of bacterial growth was recorded in the case of *S. enteritidis* and alkali extracts of *Trametes versicolor* and *Picipes badius* (2.5 mg/mL), while the water extract of *Cratherellus cornucopioides* had the lowest MIC against *L. monocytogenes* (2.5 mg/mL). Extracts derived from polypore species were not more effective than those extracted from the other groups.

### 2.2. Antiadhesion and Antibiofilm-Forming Activity

Mushroom extracts were able to prevent both bacterial strains’ adhesion to a polystyrene surface. The effect was stronger against *L. monocytogenes*, usually between 50 and 60% (Appendix A). The type of extraction was also significant in several cases where alkali extraction usually gave around 10% better results when applied in the presence of Gram-positive bacteria. In the case of *S. enteritidis*, the % of adhesion inhibition mostly went up to 30. The significant effect of mushroom species was not observed in this case. The effect was concentration-dependent. However, the dependency was not regular; lower concentrations, 0.312–0.625 mg/mL, were the most effective. In general, all extracts had strong activity when incubation took a longer time (48 h), so they expressed biofilm formation inhibition, while antiadhesion activity evolved; approximately 50% antibiofilm-forming effect went up to approximately 100%. Gram-positive bacteria were again slightly more sensitive. Alkali extracts (over 90% of samples) mostly had higher antibiofilm-forming activity, and the effect was concentration-dependent in the same manner as with the antiadhesion activity. The highest tested concentration was the least effective, or even had no effect, while the concentrations in the range of 0.312–0.625 mg/mL had the strongest antibiofilm-forming effect. 

### 2.3. Artificial Neural Network Model

With an intention to study the non-linear relationship between the anti-adhesion and anti-biofilm-forming effects of 40 extracts from 20 mushroom species against two very important food-borne bacterial species, the ANN technique was used in building the models. The ANN structure (including the biases and weight coefficients) and results depend on the initial presumptions of the matrix parameters, which are vital for ANN building and fitting to experimental data. In addition, the number of neurons in the hidden layer can change the conduct of the ANN model. In order to avoid this issue, each topology was run 100,000 times to avoid random correlation due to initial assumption and random initialization of the weights. According to this approach, the highest r^2^ value throughout the training cycle was obtained when the nine hidden neurons were used to build the ANN model (Figure 1a). 

The model was trained for 100 epochs, and the training results, i.e., train accuracy and error (loss), are presented in Figure 1b. The training accuracy increased with the number of training cycle increments until the 70–80th epoch, when it reached almost constant value. The highest train accuracy and lowest train loss were detected for the 70–80th epoch, after which a slight train accuracy increase and a train loss decrease were observed due to overfitting. More than 80 epochs for training would possibly cause high overfitting, and 70 epochs would be enough to achieve high model accuracy without any risk of overfitting (Figure 1b). 

## 3. Discussion

### 3.1. Antimicrobial and Antibiofilm Activity

The antibacterial activity of mushroom-derived compounds or mushroom extracts has been extensively explored and reported for more than two decades. It was a step towards the search for naturally derived antibiotics due to the ever-more present threat of bacterial resistance. As stated by the World Health Organization (WHO) and CDC, microbial resistance presents a global health, development, and economic threat [31,32]. Alves proposed mushroom extracts as an alternative source of new antimicrobials [33]. The authors screened 13 wild mushroom species prepared as aqueous methanolic extracts against clinical isolates of multi-resistant bacteria. Several species turned out to be especially effective, with MIC in the range of 5–20 mg/mL (similar to this study, Table 1). Extracts were more effective against Gram-positive bacteria in which case microbicidal activity was reported, while in the case of Gram-negative species, extracts were bacteriostatic. The same was observed in this and several other studies [34,35,36]. Phenolic compounds and flavonoids have been marked as the molecules responsible for mushrooms’ antimicrobial activity [37]. It has been reported that *Fomitopsis pinicola* possesses several compounds that strongly affect multidrug-resistant bacteria [36]. The same species tested here acted as an inhibitor of bacterial growth, and the alkali extract was more effective (MIC for both tested bacteria was 10 mg/mL). However, it was not the most potent among all 80 tested samples, probably due to the different solvents and bacterial strains used in this study. Concerning the method of extraction, it has been reported that methanol, ethyl acetate, and aqueous extracts accounted for 92.8% of the assays with antibacterial activity [35]. In the same study, aqueous extracts of *Clitocybe geotropa*, together with *Lentinula edodes*, had the highest antimicrobial activity against several tested pathogens. In this study, *C. geotropa* was among the least effective species (both types of extracts), which might be the consequence of the more aggressive extraction procedure applied here, which included high temperature and high pressure. In addition, alkali extraction is an efficient method for cell wall degradation, followed by the transformation of water-insoluble compounds [38]. Thus, the alkali samples were more water soluble and expressed higher antimicrobial activity. The type of solvent applied for mushroom extraction affects the type and concentration of polyphenols, and it has been recently reviewed [39]. For example, the concentration of polyphenols in turkey tail mushrooms extracted with water, ethanol, and methanol differed significantly, with the highest amount measured in the water extract. However, the highest amount of single polyphenol, p-hydroxybenzoic acid, has been confirmed in methanol extract. The overall antibacterial effect was not so strong, while almost all samples exhibited very strong (over 50%) antibiofilm activity. This is not unfounded because it has been proven that polysaccharides from bacteria such as *Escherichia coli* efficiently act as antibiofilm agents toward other pathogenic strains. They are defined as the key competition compounds and are even autoregulatory [40]. Because mushrooms live in a bacteria-loaded environment, such as soil and decaying plant material, it is possible that their compounds, including polysaccharides (which are dominant in the samples examined in this study), have a competitive function. In addition, the environment mushrooms populate is wet and, thus, encouraging biofilm formation [41]. Similarly, another group of researchers demonstrated that the ethanol extract of *Marasmius oreades* had a weak antimicrobial but a significant antibiofilm effect [42].

The problem of bacterial biofilms is well-known in the food industry and is usually treated by chemical and physical methods, which are expensive, hard to perform, vary in stability and thus efficiency, contribute to microbial resistance, and can even jeopardize consumer’s health [43,44]. Especially challenging is mature biofilm eradication due to the increase in the structure’s resistance; for example, fully formed biofilms are 10 to 1000 times more resistant to antibiotics than planktonic cells [45]. Among many food-borne pathogens, two species were selected, Gram-positive and Gram-negative, to cover principal differences connected with the bacterial structure. In addition, both species are often reported as the cause of food poisoning [7,46,47]. In this study, both adhesion and biofilm formation were prevented in the presence of various mushroom species extracts, and biofilm formation was almost completely stopped in several cases. Klančnik [16] tested crude and purified compounds from mushrooms as potential antibiofilm agents and found that crude extracts are more efficient. The percentage of efficiency, in this case, varied from 26.6 to 73.7, and the samples were stronger inhibitors of fungal biofilms. Here, the focus was on crude instead of more purified extracts because a synergistic effect was expected due to the wide palette of active compounds such as phenolic acids proven to be present in mushrooms [15,48]. The synergistic antibiofilm effect of mushroom extracts has also been reported and demonstrated by other authors [18]. However, it is not always the case, and the effect depends on the exact combination of phenolic acids and other compounds, as well as their concentrations. The same authors reported satisfactory inhibition of biofilm formation of different *L. edodes* extract samples, ranging from 20 to 30%, which is significantly lower than in the present study. They also established dose-dependent activity, although not in all cases. Lower doses when combined had additive outcomes, while higher doses acted antagonistically. The observed effect was attributed to the physicochemical interactions between the compounds in complex extracts. Kavita [49] discussed the antibiofilm activity of extracellular polymeric substances from *Oceanobacillus iheyensis*, noticing a dose-dependent relation up to a specific concentration of tested material. This was also observed in this study, and the cut-off value varied with different samples. In most cases, it was 0.312 or 0.625 mg/mL. Some of the species examined here have already been tested for their antibiofilm activity and reported by other authors, although the bacterial strains were not the same [50]. Their aqueous extract of *B. edulis* had high efficiency in removing the biofilms of *Staphylococcus aureus* and *E. coli* (78 and 94%, respectively), which is in agreement with the findings reported here for water extract of the same fungi species against *L. monocytogenes* and *S. enteritidis* (88 and 87%, respectively). In both studies, the antibiofilm effect of this popular edible mushroom was stronger against Gram-negative bacteria.

Several compounds might be responsible for the antibiofilm effect of mushrooms. Polysaccharides, proteins, peptides, and small water-soluble molecules, such as phenolic compounds, were identified as the responsible active agents [16]. Different authors pointed out that phenolic acids (gallic, ferulic, vanillic, malic, caffeic, and chlorogenic acid) are significant contributors to the antibiofilm activity of mushrooms [15,42]. Gallic, caffeic, and chlorogenic acids inhibit the metabolism of bacterial cells in a biofilm, leading to its disruption [44]. Another often-reported mechanism of the antibiofilm activity of mushrooms is the inhibition of quorum sensing [19,51,52]. Hereby, the focus was not on identifying the compounds responsible for the antibiofilm activity but to screen as many species as possible in order to set up and train a mathematical model capable of predicting the antibiofilm activity of mushrooms. However, previous studies showed that extracts prepared in the way described here mainly consisted of polysaccharides.

In addition, the mushroom extracts tested in this study mostly did not express bactericidal activity, while at the same time demonstrating an antibiofilm-forming effect. Thus, the prospect of building microbial resistance is non-existent or unlikely, which makes this material highly interesting for practical application. Moreover, mushroom polysaccharide extracts are water-soluble and mainly non-toxic, which is not the case with some other highly efficient antibiofilm polysaccharides, such as the ones isolated from non-pathogenic *E. coli* [20].

Although the predictive capabilities of the ANN modeling technique have been tested on different antimicrobial potential setups [26,27,28,53,54], this study enables the explication of the antiadhesion and antibiofilm-forming characteristics of mushroom extracts for the first time. The targeted physiological characteristics of bacterial cells relevant to the decreasing of food-borne infection have been observed by modest, routine, and repeatable assays. In addition, the mathematical methodology used in this study has been applied for the first time in the experimental investigation of water and alkali extracts of a high number of mushroom species.

### 3.2. ANN Model

The ANN model showed a good ability to generalize and predict experimental data. Based on its performance, the optimal number of neurons in the hidden layer for predicting the inhibition of adhesion and the inhibition of the biofilm-forming process was found to be 9 (MLP 25-9-2 network), resulting in high values of the coefficient of determination (r^2^), with an overall value of 0.856 for the training period, and low values of the sum of squares (SOS), as seen in Table 2.

Appendix A presents the elements of matrix *W*_1_ and vector *B*_1_ (presented in the bias row), and Appendix A presents the elements of matrix *W*_2_ and vector *B*_2_ (bias) for the hidden layer, for the ANN model, which were calculated using Equation (2). The performance of the ANN model was evaluated by comparing its calculated outputs to the experimental measurements, represented as the sum of the coefficient of determination (r^2^) between measured and calculated values for the inhibition of adhesion and inhibition of the biofilm-forming process. The results showed that the goodness of fit was 0.894, 0.863, and 0.887 for the training, testing, and validation steps, respectively, for the inhibition of adhesion and 0.856, 0.796, and 0.781 for the inhibition of the biofilm-forming process. The ANN models were able to predict the experimental variables with good accuracy for a wide range of process variables, as seen in Figure 2, which shows the comparison between the experimentally measured and ANN model-predicted values.

The ANN models were highly successful in predicting values that were very close to the desired values, as indicated by the high r^2^ value. This is in line with other studies in the literature that have reported similar results for the inhibition of adhesion and inhibition of biofilm-forming processes [55,56,57]. The ANN model used in this study was complex, with 254 weights and biases, due to the high nonlinearity of the system being investigated [55,58]. During the training period, the r^2^ value between the experimental and ANN model outputs was 0.865 for the inhibition of adhesion and inhibition of biofilm-forming processes.

The quality of the ANN model fit was assessed using χ^2^, MBE, RMSE, and MPE values [59], which should be as low as possible. Additionally, a residual analysis was performed to further evaluate the model. Skewness measures the deviation from normal symmetry, and Kurtosis measures the peakedness of a distribution. If the skewness or kurtosis values are clearly different from zero, then the distribution is asymmetrical or more peaked than normal, respectively. A high r^2^ value indicates that the variation was accounted for and that the data fit satisfactorily to the proposed model [60,61].

The predicted values were very close to the desired values in most cases, in terms of r^2^ value [56].

The fit of the model was evaluated using Table 3, where the aim is to have a high coefficient of determination (r^2^) value close to 1. Additionally, to achieve a good fit to the experimental values, it is desirable for the values of other tests, such as χ^2^, MBE, RMSE, and MPE, to be as low as possible [60,61].

### 3.3. Global Sensitivity Analysis-Yoon’s Interpretation Method

In this study, the relative influence of 25 input variables on the inhibition of adhesion and inhibition of the biofilm-forming process was analyzed. According to the results shown in Figure 3, the concentration of *L. monocytogenes* was found to have the greatest positive influence on the inhibition of adhesion, with a relative importance of approximately +8.72%. Conversely, the concentration of *S. enteritidis*, as well as the concentrations of extracts Fh and GI, were identified as the most negatively influential parameters on the inhibition of adhesion, with relative influences of −6.24%, −13.24%, and −10.81%, respectively.

For the inhibition of the biofilm-forming process, a positive influence was observed for the concentration of *L. monocytogenes*, as well as extracts of mushrooms As, Dq, Fp, and Sc, with relative influences of +6.90%, +6.03%, +9.07%, +7.70%, and +7.76%, respectively. The most pronounced negative influence on the inhibition of the biofilm-forming process was observed for the Conc. and Tv mushroom extracts, with relative influences reaching −10.04% and −10.61%.

## 4. Materials and Methods

### 4.1. Reagents

Ethanol (96% *w*/*v*), tryptic soy broth, 2,3,5-triphenyltetrazoliumchloride (TTC), and crystal violet solution (1%) were all obtained from Sigma Aldrich (St. Louis, MO, USA).

### 4.2. Mushroom Samples Collection and Identification

Mushroom species, 20 of them, were collected in the period of spring 2019–autumn 2020 at different locations in the Republic of Serbia. The identification of carpophores was performed by the authors (K.A., V.J., and L.V.) according to the methods of classical herbarium taxonomy; the micro- and macromorphology of collected specimens were compared to standard descriptions in the taxonomic monographs. The representative voucher specimens, including their mycelial cultures, were deposited in the specimen collection of the Institute of Food Technology and Biochemistry, University of Belgrade—Faculty of Agriculture. To prepare them for extraction, mushroom samples were brush-cleaned and air-dried (40 °C) to constant mass. After being powdered in a Cyclotec mill (Tecator, Hoganas, Sweden), 40 mesh, the samples were stored in a cool and dark place prior to analysis.

### 4.3. Preparation of Mushroom Extracts

The dried powdered mushroom was soaked in ethanol (70% *w*/*v*) and mixed in a magnetic stirrer, filtered, mixed with water, and autoclaved for 1 h (120 °C, 1.2 bar). After being cooled, the material was filtered through cotton wool, and the liquid part was evaporated on a hot plate magnetic stirrer until 90% of the starting volume had been removed. The concentrated sample was next precipitated with a double amount of ethanol and centrifuged (Eppendorf 5804R, Hamburg, Germany) for 10 min at 9000 rpm. The supernatant was discarded, while the solid part was dried at 40 °C and powdered in a mortar. Thus, the water extract was obtained. A filter cake left after the liquid part was separated after autoclaving was mixed with sodium hydroxide (1 M) and again autoclaved under the same conditions. The material was then cooled to room temperature, filtered through cotton wool, neutralized, and centrifuged for 10 min at 9000 rpm. As with the water extract, the alkali sample was precipitated overnight with ethanol at 4°. The next day, it was centrifuged, and the solid part was collected and dried at 40 °C and powdered. 

### 4.4. Bacterial Strains and Culture Conditions

The bacterial strains used in this study were Gram-positive *L. monocytogenes* ATTC 19111 and Gram-negative *S. enteritis* ATTC 13076. Both bacterial strains were obtained by aerobic subculturing in Tryptic Soy Broth (TSB), 24 h at 37 °C, before testing antimicrobial activity. The working concentration of the microbial suspension in the appropriate medium was 10^5^ CFU/mL and was adjusted with a McFarland densitometer (Biosan, Latvia). 

### 4.5. Determination of Mínimum Inhibitory Concentrations (MIC)

The MIC was determined by the broth microdilution method [62]. The MIC was determined using serial dilutions in concentrations ranging from 0.156 to 20 mg/mL. An amount of 50 µL of sample dilutions and 50 µL of test microorganisms in a TSB were added to a sterile 96-well microtiter plate (Sarstedt, Germany). After aerobic incubation for 24 h at 37 °C, TTC, an indicator of cellular respiration, was added to the wells in a final concentration of 0.05%. After 1 h it was assessed whether a pink color has appeared. The MIC is taken as the first lowest concentration at which no color developed. From all wells where the color did not appear, the sample was sub-cultured on TSA plates for 24 h at 37 °C. The MBC represents the lowest concentration that prevents any growth of the organism after its subculturing. Every experiment was performed in triplicate. 

### 4.6. Antiadhesion and Antibiofilm-Forming Assay 

To determine the antiadhesion potential of mushroom extracts, the same bacteria as for the MIC assay were used. The antiadhesion ability was determined using a procedure established previously [19]. To adjust the final concentration of the overnight bacterial culture suspension to 10^5^ CFU/mL, a McFarland densitometer (Biosan, Latvia) was used. The mushroom extract was dissolved in sterile Milli-Q (MQ) water in such a way as to obtain an initial concentration of 5 mg/mL, and then this solution was transferred to flatbottomed 96-well microtiter plates (Thermo Scientific™ Nunc™, Waltham, MA, USA). The bacterial culture (50 μL), previously prepared in sterile TSB, was added to the mushroom extracts. After 24 h of incubation, the plates were washed three times with sterile MQ water using a microplate washer (Rayto RT-3100, Rayto Life and Analytical Sciences Co. Ltd., Shenzhen, China). The samples were dried at 40 °C, and then a crystal violet solution (0.1 mL, 1%) was added to them. The dyeing lasted 15 min, so the dye was carefully washed off with tap water. To determine the inhibition of bacterial adhesion, a solution of acetic acid (33%) was added to the dried samples containing *L. monocytogenes*, while an ethanol solution (96%) was added to those with *S. enteritidis*, and absorbance was measured at a wavelength of 590 nm using a microplate reader (BioTek ELx 808, Winooski, VT, USA). The corresponding bacterial suspension in TSB was used as a control. The equation
(1)$$\left(\frac{Ac-As}{Ac}\right)\cdot 100$$
was used to determine the percentage of inhibition of bacterial adhesion (%IA) achieved in the presence of the appropriate concentration of mushroom extract. *Ac* is the absorbance of the control and *As* is the absorbance of the sample. 

The antibiofilm potential of mushroom extracts was determined using the procedure already described, with the difference that samples in microtiter plates were incubated for 48 h instead of 24 h. In addition, Equation (1) was applied. In this case, the percentage of inhibition of biofilm formation (%IB) in the presence of the appropriate concentration of the mushroom extract was determined.

### 4.7. Artificial Neural Network (ANN)

The multi-layer perceptron (MLP) model was used, which is known for its accuracy in approximating non-linear functions [63], for modeling the anti-adhesion and anti-biofilm effects of the mushroom extracts. To optimize the model, the Broyden–Fletcher–Goldfarb–Shanno (BFGS) algorithm was used, with 100,000 repeating steps. The experimental data was divided into three parts: 60% for training, 20% for cross-validation, and 20%for testing. The weights and biases for the hidden and output layers were grouped into two matrices: W_1_ and B_1_ and W_2_ and B_2_, respectively:(2)$$Y=f_{1}(W_{2}\cdot f_{2}(W_{1}\cdot X+B_{1})+B_{2})$$

Weight coefficients were determined during the ANN learning cycle, with the intention to minimize the error between network results and experimental values [56,64].

#### Global Sensitivity Analysis

The relative impact of the input parameters on the output variables was calculated using Yoon’s global sensitivity equation, which is based on the weight coefficients of the developed ANN models [65].
(3)$$RI_{ij}(\%)=\frac{\sum_{k=0}^{n}\left(w_{ik}\cdot w_{kj}\right)}{\sum_{i=0}^{m}\left|\sum_{k=0}^{n}\left(w_{ik}\cdot w_{kj}\right)\right|}\cdot 100\%$$
where *w*—weight coefficient in ANN model, *i*—input variable, *j*—output variable, *k*—hidden neuron, *n*—number of hidden neurons, *m*—number of inputs.

The numerical verification of the developed models was tested using reduced Chi-square (χ^2^), root mean square error (RMSE), coefficient of determination (r^2^), mean bias error (MBE), and mean percentage error (MPE) [59]:(4)$$\chi^{2}=\frac{\sum_{i=1}^{N}{\left(x_{\exp,i}-x_{pre,i}\right)}^{2}}{N-n}$$
(5)$$RMSE={\left[\frac{1}{N}\cdot \sum_{i=1}^{N}{\left(x_{pre,i}-x_{\exp,i}\right)}^{2}\right]}^{1/2}$$
(6)$$MBE=\frac{1}{N}\cdot \sum_{i=1}^{N}\left(x_{pre,i}-x_{\exp,i}\right)$$
(7)$$MPE=\frac{100}{N}\cdot \sum_{i=1}^{N}\left(\frac{\left|x_{pre,i}-x_{\exp,i}\right|}{x_{\exp,i}}\right)$$
where *x*_exp,_*_i_* stands for the experimental values and *x_pre,i_* are the predicted values calculated by the model; *N* and *n* are the number of observations and constants, respectively.

### 4.8. Statistical Analyses

The data were processed statistically using the software package STATISTICA 10.0 (StatSoft Inc., Tulsa, OK, USA). The obtained results are expressed as the mean value with standard deviation (SD). 

## 5. Conclusions

Discovering mushrooms as a source of novel antimicrobial agents in nature, this study offers the initial case study of their full potential in in vitro conditions against the bacterias *S. enteritidis* and *L. monocytogenes*. Namely, mushroom-derived extracts proved to be efficient food-borne pathogenic bacteria adhesion and biofilm-forming control agents, which is very important due to the unbefitting use of antibiotics and the unseemly control of human infections, which has led to the emergence of resistant bacterial species. Application of alternative sources of antimicrobials, which is proven in this study with 40 different water and alkali-based extracts of 20 different mushrooms, can lead to a decrease in the negative impact on public health and the global economy. For the first time, the ANN models were applied in the investigation of mushroom extracts as anti-biofilm and anti-adhesion agents, so these types of advanced mathematical tools were able to fit the experimental data well and successfully predict the output variables, with a good predictive capability. The overall coefficient of determination (r^2^) for the inhibition of adhesion and inhibition of the biofilm-forming process was 0.865, which indicates the possibility of using this data analysis for cost- and time-consuming experiments such as this one with a high number of samples. Overall, the natural material tested in this study has several benefits of interest to the food industry, such as solubility, efficiency, and non-toxicity, and it has no bactericidal activity, while at the same time, the proposed ANN model can successfully predict its antibiofilm behavior. The features examined and validated in the presented paper provide a promising avenue for the industrial setting.

## Figures and Tables

**Figure 1 antibiotics-12-00627-f001:**
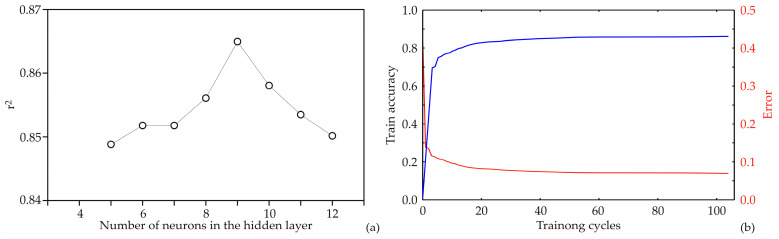
ANN calculation: (**a**) The dependence of the r^2^ value of the number of neurons in the hidden layer in the ANN model, (**b**) Training results per epoch.

**Figure 2 antibiotics-12-00627-f002:**
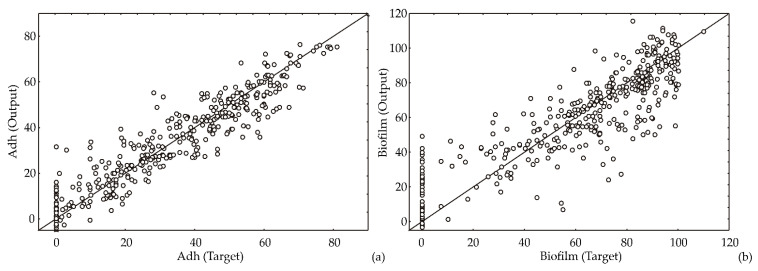
Experimental and predicted values obtained for (**a**) inhibition of adhesion and (**b**) inhibition of the biofilm-forming process.

**Figure 3 antibiotics-12-00627-f003:**
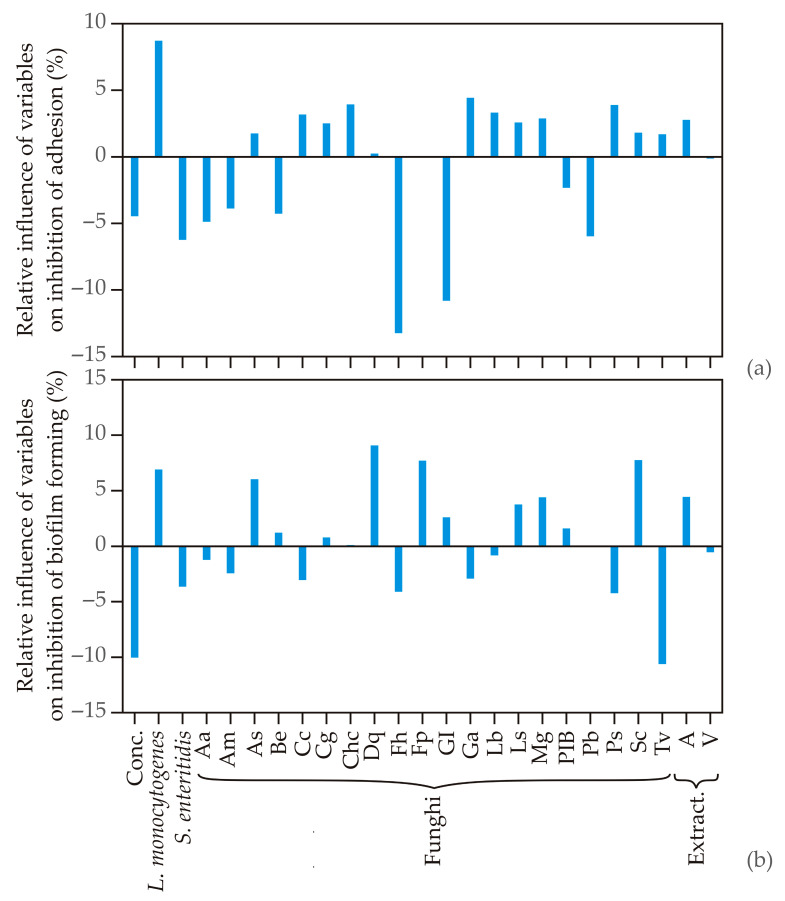
The relative importance of twenty-five input variables on (**a**) the inhibition of adhesion and (**b**) inhibition of the biofilm-forming process.

**Table 1 antibiotics-12-00627-t001:** Antimicrobial activity of mushroom extracts toward *L. monocytogenes* ATCC 19111 and *S. enteritidis* ATCC 13076, expressed as minimal inhibitory concentration (MIC) and minimal bactericidal concentration (MBC) expressed in mg/mL.

	*L. monocytogenes*		*S. enteritidis*		A Sample	*L. monocytogenes*		*S. enteritidis*	
	MIC	MBC	MIC	MBC		MIC	MBC	MIC	MBC
MgV	20	/	20	/	MgA	10	20	5	/
AsV	/	/	20	/	AsA	10	/	10	/
CcV	10	/	10	/	CcA	2.5	/	10	/
GaV	20	/	20	/	GaA	10	/	5	/
PbV	/	/	/	/	PbA	20	/	10	/
LsV	20	/	/	/	LsA	20	/	20	/
ScV	/	/	/	/	ScA	/	/	/	/
FpV	/	/	20	/	FpA	10	/	10	/
BeV	20	/	20	/	BeA	20	/	20	/
CgV	/	/	/	/	CgA	/	/	20	/
LbV	10	/	20	/	LbA	10	/	10	/
PsV	20	/	/	/	PsA	10	/	5	/
ChcV	/	/	10	/	ChcA	20	/	5	/
AaV	20	/	20	/	AaA	20	/	10	/
TvV	/	/	/	/	TvA	20	/	2.5	/
DqV	20	/	5	/	DqA	10	/	5	/
AmV	20	/	20	/	AmA	10	/	5	/
PibV	10	/	/	/	PibA	5	/	2.5	/
GlV	10	/	20	/	GlA	20	20	5	/
FhV	20	/	20	/	FhA	10	/	10	/
Gentamicin	*L. monocytogenes* MIC < 0.0024 and MBC < 0.0024 *S. enteritidis* MIC < 0.0024 and MBC < 0.0024								

All experiments were performed in triplicate. Standard deviations are not presented because all the repetitions were the same. V—water extract; A—alkali extract.

**Table 2 antibiotics-12-00627-t002:** ANN summary (performance and errors), for training, testing, and validation cycles.

Network Name	Performance			Error			Training Algorithm	Error Function	Hidden Activation	Output Activation
	Train.	Test.	Valid.	Train.	Test.	Valid.				
MLP 25-9-2	0.865	0.761	0.747	112.980	178.491	220.136	BFGS 54	SOS	Logistic	Identity

Performance term represent the coefficients of determination, while error terms indicate a lack of data for the ANN model; Train.—training process; Test.—testing process; Valid.—validation process.

**Table 3 antibiotics-12-00627-t003:** The “goodness of fit” tests for the developed ANN models.

Output Variable	χ^2^	RMSE	MBE	MPE	Skewness	Residual Analysis			
						Kurtoisis	Average	SD	Variance
Antiadhesion	59.777	7.722	−0.242	29.740	−0.454	1.373	−0.242	7.728	59.718
Antibiofilm forming	235.989	15.343	0.480	17.529	0.096	1.054	0.480	15.354	235.758

χ^2^—reduced Chi-square, RMSE—root mean square error, MBE—mean bias error, SD—standard deviation of the residuals.

## Data Availability

Not applicable.

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
