# Peer review of "Artificial Neural Network Prediction of Antiadhesion and Antibiofilm-Forming Effects of Antimicrobial Active Mushroom Extracts on Food-Borne Pathogens"

_antibiotics, 2023, doi:10.3390/antibiotics12030627_

Round 1

Reviewer 1 Report

The research is of interest to researchers and the industry. Although there has been much attention on plant bioactive ingredients and biofilm properties and antibiotics, the paper deserves to be considered for the journal.

Perhaps the authors could consider the followoing references to include in their introduction to cover this important area. 

Balaban, M., Koc, C., Sar, T. and Akbas, M.Y. (2021), Antibiofilm effects of pomegranate peel extracts against B. cereusB. subtilis, and E. faecalis. Int. J. Food Sci. Technol., 56: 4915-4924; Balaban, M., Koc, C., Sar, T. and Akbas, M.Y. (2021), Antibiofilm effects of pomegranate peel extracts against B. cereusB. subtilis, and E. faecalis. Int. J. Food Sci. Technol., 56: 4915-4924; Shevkani, K., Singh, N., Patil, C., Awasthi, A. and Paul, M. (2022), Antioxidative and antimicrobial properties of pulse proteins and their applications in gluten-free foods and sports nutrition. Int J Food Sci Technol, 57: 5571-5584.)

It would have been interesting to have isolated the bioactive ingredients from the mushrooms and prove their antibiotic effectiveness. Is there similar work that the authors could point to to illustrate this sort of relationship ?

What effect do the different solvents have on the efficacy of the ingredients? Could you please expand on this and include other research to substantiate your observations. 

Could you discuss how plant phytochemicals play an important role in biofilm control, possibly using the following reference as a comparison. Kumar, P., Mahato, D.K., Gupta, A., Pandhi, S., Mishra, S., Barua, S., Tyagi, V., Kumar, A., Kumar, M. and Kamle, M. (2022), Use of essential oils and phytochemicals against the mycotoxins producing fungi for shelf-life enhancement and food preservation. Int J Food Sci Technol, 57: 2171-2184. 

The overall coefficient was high and this shows that the model used was appropriate. Would thermal treatment of food materials affect the model and the use of the ingredients ?

Author Response

The authors would like to thank the reviewer for the effort to go through our work and help us to improve it.

  1. “Perhaps the authors could consider the followoing references to include in their introduction to cover this important area. 

Balaban, M., Koc, C., Sar, T. and Akbas, M.Y. (2021), Antibiofilm effects of pomegranate peel extracts against B. cereusB. subtilis, and E. faecalis. Int. J. Food Sci. Technol., 56: 4915-4924; Balaban, M., Koc, C., Sar, T. and Akbas, M.Y. (2021), Antibiofilm effects of pomegranate peel extracts against B. cereusB. subtilis, and E. faecalis. Int. J. Food Sci. Technol., 56: 4915-4924; Shevkani, K., Singh, N., Patil, C., Awasthi, A. and Paul, M. (2022), Antioxidative and antimicrobial properties of pulse proteins and their applications in gluten-free foods and sports nutrition. Int J Food Sci Technol, 57: 5571-5584.)”

The following is now included in the main text: Plants are already well-known and traditionally used sources of antimicrobial and antibiofilm compounds. Different parts, like leaf, root, fruit, or peel are rich in antibiofilm compounds as has been demonstrated recently using the simple and cheap crystal violet staining method (Balaban, M., Koc, C., Sar, T., Akbas, M.Y. Antibiofilm effects of pomegranate peel extracts against B. cereusB. subtilis, and E. faecalis. Int. J. Food Sci. Technol., 2021, 56, 4915-4924). Recently, mushrooms became…

  1. “It would have been interesting to have isolated the bioactive ingredients from the mushrooms and prove their antibiotic effectiveness. Is there similar work that the authors could point to to illustrate this sort of relationship ?”

The focus of this research was the antibiofilm activity of mushroom extracts since this area is scarcely researched and little is known about it. We mentioned this in lines 174-175, reference 35 (the first version of the manuscript). Also, lines 235-241.

However, as the reviewer pointed out, it is interesting to know the exact group of antimicrobial compounds too, so we would add the next: Phenolic compounds and flavonoids had been marked as responsible molecules for mushroom’s antimicrobial activity (Barros, L., Calhelha, R., Vaz, J., Ferreira, I.C.F.R., Baptista, P., Estevinho, L.M. Antimicrobial activity and bioactive compounds of Portuguese wild edible mushrooms methanolic extracts. Eur. Food Res. Technol. 2007, 225, 151–156). It has been reported that Fomitopsis pinicola

  1. “What effect do the different solvents have on the efficacy of the ingredients? Could you please expand on this and include other research to substantiate your observations.”

Yes, the reviewer observed it correctly. Solvents and the extraction procedure, besides the mushroom species, have the most prominent effect on antibiofilm and antimicrobial activity. However, we already discussed it: in lines 182-187, reference 36 (the first version of the manuscript). But we will also add: The type of solvent applied for mushroom extraction affects the type and concentration of polyphenols and it has been recently reviewed (Kozarski, M., Klaus, A., van Griensven, L., Jakovljevic, D., Todorovic, N., Wan-Mohtar, W.A.A.Q.I. Mushroom β-glucan and polyphenol formulations as natural immunity boosters and balancers: nature of the application. Food Sci. Hum. Wellness, 2023, 12, 378-396). For example, the concentration of polyphenols in turkey tail mushrooms extracted with water, ethanol and methanol differed significantly, with the highest amount measured in the water extract. However, the highest amount of single polyphenol, p-hydroxybenzoic acid has been confirmed in methanol extract. The overall antibacterial effect…

  1. “Could you discuss how plant phytochemicals play an important role in biofilm control, possibly using the following reference as a comparison. Kumar, P., Mahato, D.K., Gupta, A., Pandhi, S., Mishra, S., Barua, S., Tyagi, V., Kumar, A., Kumar, M. and Kamle, M. (2022), Use of essential oils and phytochemicals against the mycotoxins producing fungi for shelf-life enhancement and food preservation. Int J Food Sci Technol, 57: 2171-2184 “

We acknowledge that plant biomolecules are powerful antibiofilm agents, however, the material we worked with here belongs to a different kingdom. The chemical nature, methods of extraction and mechanisms of action of plants and fungi are different and we did not want to burden our already long manuscript with not directly related materials.

  1. “The overall coefficient was high and this shows that the model used was appropriate. Would thermal treatment of food materials affect the model and the use of the ingredients?”

If the reviewer is pointing to thermal treatment in food industry systems possibly coated with fungal extracts and the food passing through it-we can not know for sure what will happen since that kind of experiment hasn’t been done so far. We have done the pioneering work by modeling the possible effects. However, the next step is to get to the point of this type of material approval, which might include laminar and turbulent flow simulators, and finally check it under real industrial conditions. At this moment we can only speculate that some compounds like phenolics will be degraded due to high temperatures while polysaccharides (the main compound of two types of extracts examined here) are stable up to 350ËšC. This guarantees that a significant portion of activity will be preserved.

Reviewer 2 Report

In the methodology, it is not specified which mushrooms were used, where they were collected or how they were obtained. In the same way, the process of extracting the mushrooms is not specified. These aspects are extremely important to be able to understand the results obtained and, at some given moment, that they can be replicated.

In addition, the results section does not detail the results of the neural network, which is one of the main objectives of the work.

On the other hand, in some paragraphs misspelled scientific names appear, that is, they are not in italics.

Author Response

  1. “In the methodology, it is not specified which mushrooms were used, where they were collected or how they were obtained. In the same way, the process of extracting the mushrooms is not specified. These aspects are extremely important to be able to understand the results obtained and, at some given moment, that they can be replicated.”

The authors would like to thank the reviewer for the comments. Indeed, we did not include the section about mushroom material identification, however, the list of mushrooms is long and we did not want to burden the main text so the species names were provided as supplementary material. The detailed method of mushroom extract preparation was already explained in section 4.2.

As for the places of mushroom collection, identification and specimen deposition, we are now adding section 4.2. Mushroom samples collection and identification

Mushroom species, 20 of them, were collected in the period of spring 2019- autumn 2020 at different locations in the Republic of Serbia. The identification of carpophores was performed by authors (K.A., V.J., and L.V.) according to the methods of classical herbarium taxonomy, micro- and macromorphology of collected specimens were compared to standard descriptions in the taxonomic monographs. The representative voucher specimens including their mycelial cultures were deposited in the specimen collection of the Institute of Food Technology and Biochemistry, University of Belgrade - Faculty of Agriculture. To prepare them for extraction mushroom samples were brush-cleaned and air-dried (40°C) to constant mass. After being powdered in a Cyclotec mill (Tecator, Hoganas, Sweden), 40 mesh, the samples were stored in a cool and dark place prior to analysis.

  1. “In addition, the results section does not detail the results of the neural network, which is one of the main objectives of the work.”

Thank you very much for this observation, section 2.3. Artificial neural network model was added to the text, according to the Reviewer’s comment.

  1. “On the other hand, in some paragraphs misspelled scientific names appear, that is, they are not in italics.”

Thank you very much, all microorganisms’ names are now in italics.

Reviewer 3 Report

The manuscript reports antibacterial, anti-adhesion and anti-biofilm forming effects of extracts from mushrooms against Listeria monocytogenes and Salmonella enteritidis. Because of methodological flaws, I do not recommend the manuscript for publication.

The weaknesses of the manuscript are as follows:

The details on the person responsible for taxonomical identification and authentication of mushroom species, voucher specimen numbers (VSNs) and name of place were VSNs are deposited are not provided. Data on traditional uses of mushroom species assayed would also be interesting to potential readers.

The methodology of antimicrobial susceptibility testing should follow standard methods (e.g. CLSI, 2009; FDA, 2018; ISO, 2019).

In general, the concentrations of extract tested in biological assays are too high. For example, according to the recommendations previously proposed for more effective assessment of antimicrobial potential of natural products, minimum inhibitory concentration (MIC) values below 100 μg/ml for mixtures (extracts) should be considered as promising activity/highly effective. In addition, samples with MICs higher than 1 000 μg/ml should strictly be evaluated as no active (Kokoska et al., 2019). Based on these criteria, the results achieved for extracts assayed in the study (MICs ranging from 2 500 to 20 000 μg/ml) cannot be considered as effective concentrations.

Positive antibiotic control has not been included in antimicrobial assays.

References:

CLSI 2012. Methods for Dilution Antimicrobial Susceptibility Tests for Bacteria That Grow Aerobically; Approved Standard-Ninth Edition. CLSI document M07-A9. Wayne, PA, USA: Clinical and Laboratory Standards Institute.

FDA Guidance 2018. Class II Special Controls Guidance Document: Antimicrobial Susceptibility Test Systems.

ISO 2019. Susceptibility Testing of Infectious Agents and Evaluation of Performance of Antimicrobial Susceptibility Devices, Part 1. Broth micro-dilution reference method for testing the in vitro activity of antimicrobial agents against rapidly growing aerobic bacteria involved in infectious diseases. Second edition. ISO/DIS 20776-1, Geneva, Switzerland.

Kokoska L, Kloucek P, Leuner O, Novy P. 2019. Plant-derived products as antibacterial and antifungal agents in human health care. Current Medicinal Chemistry, 26(29): 5501-5541.

Author Response

“The manuscript reports antibacterial, anti-adhesion and anti-biofilm forming effects of extracts from mushrooms against Listeria monocytogenes and Salmonella enteritidis. Because of methodological flaws, I do not recommend the manuscript for publication.” 

The weaknesses of the manuscript are as follows:

  1. “The details on the person responsible for taxonomical identification and authentication of mushroom species, voucher specimen numbers (VSNs) and name of place were VSNs are deposited are not provided.”

The authors would like to thank the reviewer for this comment. Indeed, we did not include this important section in order not to burden the text. However, we now added the necessary details. 4.2. Mushroom samples collection and identification

Mushroom species, 20 of them, were collected in the period of spring 2019- autumn 2020 at different locations in the Republic of Serbia. The identification of carpophores was performed by authors (K.A., V.J., and L.V.) according to the methods of classical herbarium taxonomy, micro- and macromorphology of collected specimens were compared to standard descriptions in the taxonomic monographs. The representative voucher specimens including their mycelial cultures were deposited in the specimen collection of the Institute of Food Technology and Biochemistry, University of Belgrade - Faculty of Agriculture. To prepare them for extraction mushroom samples were brush-cleaned and air-dried (40°C) to constant mass. After being powdered in a Cyclotec mill (Tecator, Hoganas, Sweden), 40 mesh, the samples were stored in a cool and dark place prior to analysis.

  1. “Data on traditional uses of mushroom species assayed would also be interesting to potential readers.”

Indeed, this information can be interesting. However, the focus of this work was not an ethnomycological aspect which might divert readers from the practical industrial aspect and model development, away from mushroom traditional uses.

  1. “The methodology of antimicrobial susceptibility testing should follow standard methods (e.g. CLSI, 2009; FDA, 2018; ISO, 2019).”

Although we acknowledge that standard methods can not be doubted the work presented here was done under “research conditions” following the literature dealing with the same topic, also cited in this manuscript. For example, well-cited work examining mushroom antimicrobial activity conducted by Barros, L., Calhelha, R., Vaz, J., Ferreira, I.C.F.R., Baptista, P., Estevinho, L.M. Antimicrobial activity and bioactive compounds of Portuguese wild edible mushrooms methanolic extracts. Eur. Food Res. Technol. 2007, 225, 151–156, or Sharma S, Prakash S. To detect the minimum inhibitory concentration and time-kill curve of shiitake mushroom on periodontal pathogens: An in vitro study. J. Indian Soc. Periodontol. 2019, 23, 216-219. used the same method (Minimum Inhibitory Concentration) we applied.

  1. In general, the concentrations of extract tested in biological assays are too high. For example, according to the recommendations previously proposed for more effective assessment of antimicrobial potential of natural products, minimum inhibitory concentration (MIC) values below 100 μg/ml for mixtures (extracts) should be considered as promising activity/highly effective. In addition, samples with MICs higher than 1 000 μg/ml should strictly be evaluated as no active (Kokoska et al., 2019). Based on these criteria, the results achieved for extracts assayed in the study (MICs ranging from 2 500 to 20 000 μg/ml) cannot be considered as effective concentrations.

The authors agree with the reviewer’s observation. However, it is true for plant-derived compounds. In the case of mushrooms, effective concentrations are much higher and are selected based on the literature available data, which we cited in our work and also compared our results with (for example, reference 32). Also, the goal was not to search for green compounds with antibacterial but antibiofilm activity. The classical analysis procedure (presented in the literature) requires that first antibacterial activity needs to be tested and MIC identified, in cases where they exist. Next, antibiofilm activity is tested. As we already stated in the Introduction and Discussion section (lines 62-69 and 248-250, the first version of the manuscript) materials used in the food industry and medicine showing antibiofilm effect ideally should not exhibit antibacterial activity. The reason is to avoid building up microbial resistance.

  1. “Positive antibiotic control has not been included in antimicrobial assays.”

Gentamicin was used as positive control and it is included in Table 1.

Round 2

Reviewer 3 Report

Because the authors did not satisfactory address my comments, I do not recommend manuscript for publication.